# Structural vulnerability to narcotics-driven firearm violence: An ethnographic and epidemiological study of Philadelphia's Puerto Rican inner-city

Joseph Friedman[1]*, George Karandinos[2], Laurie Kain Hart[3], Fernando Montero Castrillo[4], Nicholas Graetz[5], Philippe Bourgois[1]*

1 Center for Social Medicine and Humanities, University of California, Los Angeles, CA, United States of America, 2 Harvard Medical School, Boston, MA, United States of America, 3 Department of Anthropology, University of California, Los Angeles, CA, United States of America, 4 Department of Anthropology, Columbia University, New York, NY, United States of America, 5 Department of Demography, University of Pennsylvania, Philadelphia, PA, United States of America

* pbourgois@gmail.com (PB); joseph.robert.friedman@gmail.com (JF)

**Data Availability Statement:** All quantitative data used in this analysis are publicly available. Crime data are available at: https://www.opendataphilly.

## Abstract

### Background

The United States is experiencing a continuing crisis of gun violence, and economically marginalized and racially segregated inner-city areas are among the most affected. To decrease this violence, public health interventions must engage with the complex social factors and structural drivers—especially with regard to the clandestine sale of narcotics—that have turned the neighborhood streets of specific vulnerable subgroups into concrete killing fields. Here we present a mixed-methods ethnographic and epidemiological assessment of narcotics-driven firearm violence in Philadelphia's impoverished, majority Puerto Rican neighborhoods.

### Methods

Using an exploratory sequential study design, we formulated hypotheses about ethnic/racial vulnerability to violence, based on half a dozen years of intensive participant-observation ethnographic fieldwork. We subsequently tested them statistically, by combining geo-referenced incidents of narcotics- and firearm-related crime from the Philadelphia police department with census information representing race and poverty levels. We explored the racialized relationships between poverty, narcotics, and violence, melding ethnography, graphing, and Poisson regression.

### Findings

Even controlling for poverty levels, impoverished majority-Puerto Rican areas in Philadelphia are exposed to significantly higher levels of gun violence than majority-white or black neighborhoods. Our mixed methods data suggest that this reflects the unique social position of these neighborhoods as a racial meeting ground in deeply segregated Philadelphia, which has converted them into a retail endpoint for the sale of astronomical levels of narcotics.

org/dataset/crime-incidents. Sociodemographic census tract data are available at: https://factfinder.census.gov/faces/nav/jsf/pages/index.xhtml.

**Funding:** The original ethnographic fieldwork was supported by National Institutes of Health grant DA010164 with background/comparative data facilitated by DA27599, UL1TR001881. The funder had no role in study design, data collection and analysis, decision to publish, or preparation of the manuscript.

**Competing interests:** Dr. Philippe Bourgois was a Guest Editor for the Substance Use, Misuse and Dependence: Prevention and Treatment special collection, however, he played no role in reviewing this manuscript. No other competing interests exist.

## Implications

We document racial/ethnic and economic disparities in exposure to firearm violence and contextualize them ethnographically in the lived experience of community members. The exceptionally concentrated and high-volume retail narcotics trade, and the violence it generates in Philadelphia's poor Puerto Rican neighborhoods, reflect unique structural vulnerability and cultural factors. For most young people in these areas, the narcotics economy is the most readily accessible form of employment and social mobility. The performance of violence is an implicit part of survival in these lucrative, illegal narcotics markets, as well as in the overcrowded jails and prisons through which entry-level sellers cycle chronically. To address the structural drivers of violence, an inner-city Marshall Plan is needed that should include well-funded formal employment programs, gun control, re-training police officers to curb the routinization of brutality, reform of criminal justice to prioritize rehabilitation over punishment, and decriminalization of narcotics possession and low-level sales.

## Introduction

*It wasn't even supposed to happen like that. I was gonna smack him . . .but he kept talking. I wasn't even gonna shoot him, but it just happened too fast man. I don't know . . . This the dumbest thing I ever did in my life. I just don't want to go back to the same nut shit when I get home. Philly is like the fuckin' devil. I need to figure out a game plan to keep me away from the streets. I need to have a job before I get out of here. And I don't know how that's goin' to work. I ain't never had no job before."*

-18-year-old Leo, recently sentenced to 10 years in prison for shooting another young man in a narcotics-related disagreement, just months after he first gains lucrative employment in the narcotics economy of inner-city Philadelphia.

In 2007, our ethnographic team set out to study urban poverty in a majority Puerto-Rican neighborhood in North Philadelphia. We immersed ourselves for six years in Philadelphia's sprawling open-air narcotics market located in the heart of the city's Puerto Rican area. We rented an apartment on a block surrounded by multiple heroin and cocaine sales points, and two members of our team (George Karandinos and Fernando Montero) lived there full-time, directly observing the sale of massive retail quantities of heroin and cocaine and participating in the routine activities of daily life in the neighborhood. Fig 1, part A shows the field site in North Philadelphia.

One of the most immediately salient features of the neighborhood was the pervasiveness of violence. As we tape-recorded interviews with residents, seeking to understand their lives, the sound of gunshots regularly punctuated the flow of conversation. On nearly every block in the area, at least one homicide occurred during the years we studied this neighborhood (Fig 1, part B). In this small, approximately 10 square block area, it was rare for a month to pass without a homicide (Fig 1, part C), and at least half a dozen armed assaults and robberies. The sound of gunshots on the blocks surrounding us became a fact of everyday life.

This level of persistent violence is not unique to inner-city Puerto Rican Philadelphia. The United States is broadly facing a continuing crisis of gun violence, and economically marginalized inner-city communities of color are among the most affected by homicide [1–6] even as national levels of homicide decreased during our fieldwork years. Levels of interpersonal

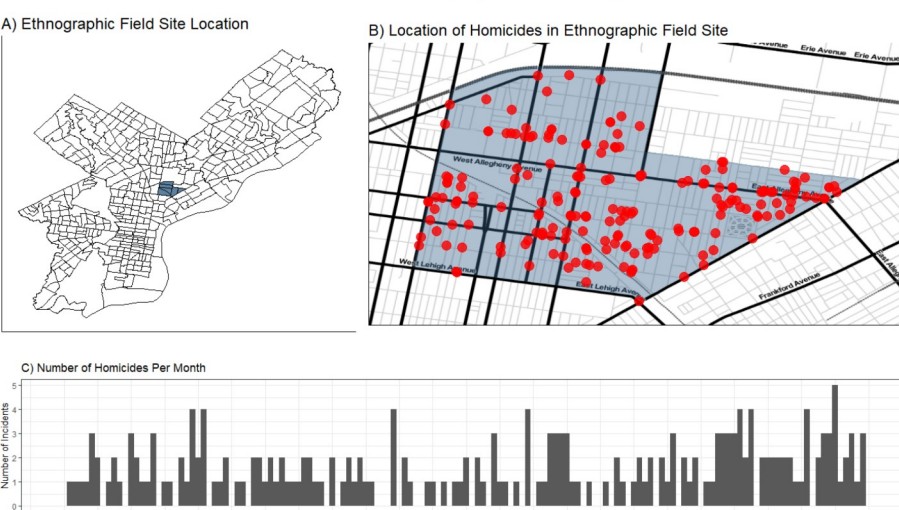

**Fig 1. Homicides in ethnographic field site from 2006–2017 (203 total).** A) Location of ethnographic field site in North Philadelphia. B) Location of each homicide incident occurring in the ethnographic site during this time. C) Monthly trends in the number of homicides occurring in the field site.

violence are higher in the US than in any other developed nation, and guns are implicated in a disproportionate percent of the associated mortality, morbidity, and costs to society [7–9]. There is a growing academic and political understanding that this crisis must be approached in a systemic, public-health oriented fashion [10]. The traditional response focusing on modifying individual factors—such as mental health interventions or personal gun safety practices—has done little to curb violence, failing to address the deep roots of the crisis and its maladministration by law enforcement, the courts, and corrections [2]. It is becoming increasingly clear that to successfully reduce firearm violence, public health interventions must engage with the complex social factors and structural drivers that undergird it. A number of programs have been able to make inroads with violence reduction among inner-city populations in a cost-effective manner through community and hospital-based interventions [11–15]. Despite these sporadic efforts, firearm homicide remains pervasive across inner-city America, with casualties largely falling along lines of race and class. This reflects the deeply entrenched systemic nature of firearm violence in inner-city America which overwhelming claims the lives of the most structurally vulnerable members of society [16].

Early in the course of our fieldwork we realized that much of the burden of the gun violence occurring around us was not random or totally senseless. It was often tied directly to interpersonal power dynamics, petty insults, and narcotics business negotiations regarding unpaid debts, the enforcement of monopoly control, or the theft of drugs. The killings that periodically took place, therefore, could not be averted with simple programs seeking to encourage teens to think twice before purchasing an illicit firearm or manage their anger more responsibly. Violence in our neighborhood was a constant anxiety because of the economic and political forces that converted this economically-abandoned industrial neighborhood into a hub of competing open-air narcotics sales points.

Our neighbors were rendered vulnerable to violence by structural forces [16–19], such as their low educational levels, marginalization from legal employment opportunities, easy access to wholesale supplies of heroin and cocaine, easy access to licensed and unlicensed firearms, and alienation from social services. Framed in classic public health terminology, each of these

circumstances could be considered a social determinant of health—a social factor, out of the individual's immediate control that modifies their personal risk for adverse health outcomes, such as addiction or death by homicide [20]. Employing the concept of structural vulnerability, however, we can go beyond isolated risk factors to assess how each person's risk relates to their specific position in the social and economic hierarchies of their local environment [21]. Structural vulnerability is driven by power relations, large-scale social and economic forces that marginalize certain individuals and cause and exacerbate, their illnesses. Beyond just macro-level economic disparities, structural vulnerability also engenders an understanding of how local cultural and social dynamics often mediate or exacerbate personal risk of illness. For example, in our field site we saw how community dynamics and cultural patterns regulate and place limits on the violence occurring in the community. Therefore, any intervention seeking to address violence in inner-city Puerto Rican Philadelphia must be both structurally and culturally competent—it must understand the specific position of community members relative to social factors and structural drivers—if it is to be effective in truly reducing the level of mortality.

In this study, we seek to characterize the structural vulnerability of the Puerto Rican community in Philadelphia to gun violence, using a mixed-methods approach. We build on our team's previous collaborative efforts to put ethnographic observations from the inner-city into direct conversation with epidemiological data [22–24], and aim to present an assessment of firearm violence in these communities that is quantitatively rigorous with regard to the larger picture while also conveying the lived experience of the human suffering reflected in the macro statistics.

## Methods

We used an exploratory sequential [25] mixed methods [26,27] study design. In this approach, the first stage entailed an exploratory, ethnographic process by which the ethnographic team deeply acquainted themselves with the field site and the relevant social dynamics. In the second stage, ethnographic data were analyzed using formal qualitative methods. Qualitative conclusions were then used to generate hypotheses about macro-level dynamics that could be tested quantitatively. Epidemiological methods were then applied to quantitative data, in order to assess the hypothesis that had been initially generated ethnographically. Iteratively, qualitative and quantitative results were compared, and placed into dialogue with each other, to produce final conclusions.

### Ethnographic methods

The members of the ethnographic team (FM,GK,LH,PB) immersed themselves for six years in the heart of Philadelphia's sprawling open-air narcotics markets located in the core of the city's Puerto Rican area, from Fall 2007 through Summer 2013 with strategic follow-up work through Spring 2019 [24,28]. We rented an apartment in the field site in a decaying subdivided row home where GK and FM lived full-time and LH and PB visited regularly, sometimes staying the night, enabling us to participate in the local social scene as neighbors. With IRB approval, we conducted interviews in a conversational format while accompanying respondents in their daily lives and activities. Through the standard anthropological methods of participant-observation and immersion in the social environment, we were able to document hard-to-observe, stigmatized, and illegal activities [29]. Our long-term immersion reduced social desirability bias and also enabled us to compare the reported practices of respondents with direct observation of their activities and emotions. Over the course of these years we developed genuine close friendships with many of our respondents, which we still enjoy.

These warm relationships facilitate long-term follow-up with hard-to-reach criminal justice- and substance use disorder-impacted, vulnerable populations [24,29,30].We watched young neighbors come of age and climb the ranks of the narcotics economy, accompanied them through the criminal justice system, visited them in jail, and corresponded with them in prison [31].

These ethnographic experiences yielded over 1300 pages of field notes, and nearly 2000 pages of transcripts from hundreds of interviews with dozens of respondents, hundreds of photographs, and dozens of hours of video. Our final qualitative database consisted of 3,189 pages of fieldnotes and transcriptions of interviews, representing 471 separate files, which often includes data from multiple participants and overlapping interviews. Our data included detailed documentation of both firearm and barehanded violence, police corruption and brutality, the logistics of narcotics sales, friend and family relationships, and interactions between members of our neighborhood with state institutions, especially schools, clinics, courts and carceral facilities. All names describing research participants in this work are pseudonyms. We obtained verbal informed consent because for ethnographic data, the benefits to protecting confidentiality by documenting it with a signature are outweighed by the risk that a hard copy paper signature represents as the most directly-identifying legal marker of the participant. Verbal informed consent was also obtained from parents or guardians for unemancipated minors as well as from the minors themselves. For additional methodological details we refer to more extensive, previously published discussions of our ethnographic methodology [23,24,28,29,32,33].

All ethnographic data was loaded into the NVivo qualitative analysis platform and coded for themes and characters. Analysis was done iteratively, with preliminary rounds beginning while the fieldwork was still being conducted, which was used to prompt follow-up assessments and refine research questions. Self-reports of behavior were triangulated against qualitative observations of practices occurring in real time in the natural environment. As we began the preliminary analysis, we also further explored the generalizability of our observations in the later stages of our fieldwork. This was accomplished by strategically following up with individuals who had made statements or exhibited behaviors counter to the dominant ideas in the community about research questions. Through long term documentation, we were able to assess whether these "exceptions prove the rule" or if they represented a significant divergence from our findings that required us to shift our analysis. Ultimately, what we present here is a reflection of the dominant social dynamics and views in each neighborhood—those that might have the most impact on the macro-level statistical trends we observe. Final rounds of qualitative analysis occurred until "saturation" had been achieved, at which point the qualitative analysis yielded several quantitative hypotheses about the racial/ethnic and social dynamics of firearm violence in Philadelphia.

We tested statistically our main hypothesis that, even controlling for poverty levels, poor majority-Puerto Rican neighborhoods in Philadelphia experience higher levels of violence and narcotics traffic than white or black areas. We also sought to confirm our qualitative understandings of unique properties of the field site, in terms of sociodemographic trends including the degree of marginalization, as well as the uniqueness levels of narcotics and violence-related crime experienced in our community. Quantitative data were sought, where possible, to confirm or refute the understandings we had built ethnographically.

## Quantitative methods

Data describing levels of narcotics- and violence-related crime was obtained from files made publicly-available by the Philadelphia police department [34]. The database describes nearly 3

million geo-coded and time-stamped incidents that occurred from 2006 to 2017, from which we selected the approximately 200,000 observations describing crime incidents in the "Aggravated Assault Firearm", "Narcotic / Drug Law Violations", "Weapon Violations", or "Homicide–Criminal" categories. We used the latitude and longitude of each event to assign it to a census tract. We obtained census tract level information representing ethnic, racial, and poverty data for the pooled 2012–2016 period from the American Communities Survey (ACS) [35]. We identified the majority social group in each census tract from these files, as either "black", "white", "Puerto Rican", or "Non-Puerto Rican Hispanic". We used "non-Hispanic White" status as "white" in this study. Data were mapped using a census-tract shapefile made available by the city of Philadelphia[36].

We explored the relationship between neighborhood ethnic composition, levels of violence and narcotics-related incidents, and poverty, using mapping, plotting, and Poisson regression. To test if rates of violence and narcotics are higher in Puerto Rican areas, controlling for poverty, we use Poisson regression to estimate the relationship shown in Eq 1.

$$Crime_{T,G} = +\beta_1 \cdot I_{Wh} + \beta_2 \cdot I_{PR} + \beta_3 \cdot I_{Bl} + \beta_4 \cdot Pov \cdot I_{Wh} + \beta_5 \cdot Pov \cdot I_{PR} + \beta_6 \cdot Pov \cdot I_{Bl} \quad (1)$$

Where $Crime_{T,G}$ represents the tract-level counts of violence- or narcotics-related crime of type $T$ for tracts of majority social group $G$ occurring between 2006 and 2017, $Pov$ represents the percent of the population living under the poverty line during the 2012 to 2016 period, and $I_{Wh}$, $I_{PR}$, and $I_{Bl}$ are binary indicators for majority-white, Puerto Rican, and black neighborhoods respectively. This model estimates separately the poverty-crime gradient for each social group with distinct slopes and intercepts. The small number of majority non-Puerto Rican Hispanic neighborhoods were not included in this analysis due to insufficient sample size. We also excluded 4 majority-white census tracts that appear to have high levels of poverty because they are almost entirely composed of relatively high socio-economic status college students. This regression was repeated separately for each $T$ type of crime including "Aggravated Assault Firearm", "Weapon Violations", "Homicide–Criminal," and "Narcotic/Drug Law Violations."

In order to test our hypothesis that the most impoverished Puerto Rican neighborhoods have higher levels of violence and narcotics traffic relative to majority-black or majority-white areas, even controlling for poverty, we predicted $Crime_{T,G}$ values for each social group and crime outcome modeled where $Pov = 60\%$, which is similar to the neighborhood in which we conducted field research. We then calculated the ratio between the estimated counts of violence and narcotics related crime for each social group (e.g. the ratio of majority-Puerto Rican to majority-black rates of homicides at 60% poverty) and calculated the 95% prediction interval for each ratio. We modeled total counts of narcotics or violence related crime, without using an offset, in order to measure neighborhood-level effects. Given the dynamics of narcotics-related violence, the individuals who are perpetrating or suffering from violence in a given location often don't live in that census tract. By using total counts, we measure the cumulative exposure to violence experienced by people living in a certain census tract with a given set of social characteristics. All analyses were conducted using R version 3.5.1 [37]. All data used in this analysis are publicly available [34,35]. All code used in this analysis are available in the S1 Appendix.

In the last stage of analysis, the qualitative and quantitative results were synthesized to produce final conclusions. This entailed the comparison of qualitative results and qualitatively-generated hypotheses with quantitative data and results. Whenever possible, qualitative and quantitative data describing similar phenomena were placed in close proximity to one another; e.g. maps describing socioeconomic deprivation placed adjacent to ethnographic descriptions of the same phenomena. Final conclusions were drawn after considering all data, and via

numerous rounds of consensus-driven feedback from the ethnographic and quantitative analysis teams.

## Results and discussion

### Life in the deindustrialized and hyper-segregated inner-city

Leo, the young man featured in the opening lines of this article, was born into the poorest corner of Philadelphia's Puerto Rican inner-city, once the core of Philadelphia's 19th century industrial sector. His grandparents came to the mainland United States, alongside millions of other Puerto Ricans in the 1950s and 1960s, seeking factory employment. Unfortunately, Puerto Ricans began migrating to Philadelphia just as the manufacturing industry started shrinking in size. By the 1980s the neighborhood was characterized by abandoned factories, empty row homes, vacant lots, and piles of rubble [4]. Philadelphia has not recovered from deindustrialization, and it remains the poorest of the largest ten American cities [28]. It experienced the classic US pattern of "white flight" to the suburbs, with the creation of hyper-segregated, poor, racialized ghettos with high vacancy rates in its former factory neighborhoods. This infrastructural decay represents a perfect environment for harboring difficult-to-police drug markets, sex work, drug consumption shooting galleries, and homeless squats [16]. Fig 2 demonstrates the marked segregation along racial lines in Philadelphia, as well as the correlation with poverty, narcotics, and firearm and violence rates (see S1 Fig for a continuous color scale). The majority-Puerto Rican section is wedged between hyper-segregated white and black neighborhoods and is traversed by a subway system and two major interstate highways, facilitating access for multi-racial urban and suburban customers from the wider region, including New Jersey, Delaware, Pennsylvania, and Maryland [24,38].

The Puerto Rican inner-city has economically fared worse than the city average, experiencing extreme levels of private and public sector disinvestment, worsened by neoliberal policies that slashed the social safety net system [39]. As a result, deeply entrenched poverty has prevailed. According to ACS data ~60% of households in the neighborhood had incomes under the poverty line between 2012 and 2016 (Fig 2). During our fieldwork there were virtually no legal businesses in the neighborhood offering meaningful employment opportunities. Quite literally in the shadow of abandoned factories, the narcotics economy rose up to fill the employment vacuum (Fig 3).

### The narcotics economy in Puerto Rican Philadelphia

For Leo, a low-income male high school dropout, the narcotics economy represented the only "equal opportunity" employer available to him. An enterprising young man, Leo was motivated to succeed. He had no local examples of successful male role-models in his social network who achieved economic stability and social mobility without participating in the retail sale of narcotics. Furthermore, all of Leo's immediate family and almost all of the young males in his extended social network were either employed selling heroin and cocaine, incarcerated, or prematurely deceased. A few older males and recent immigrants from Puerto Rico in the neighborhood did manage to find work in downtown office complexes as janitors, or commuted to jobs in suburban warehouses, golf courses, or slaughterhouses. There they competed with undocumented Dominican, Mexican and Central American day laborers for minimum wage employment, with almost no prospects for upward mobility or promotion. Leo did, in fact, briefly seek similar employment as a janitor, and even asked several members of the ethnographic team to serve as references for his application. However, he quickly abandoned this pursuit after several applications submitted for minimum-wage positions resulted in zero calls or interviews. In contrast, several young men whom Leo admired, including his older brother

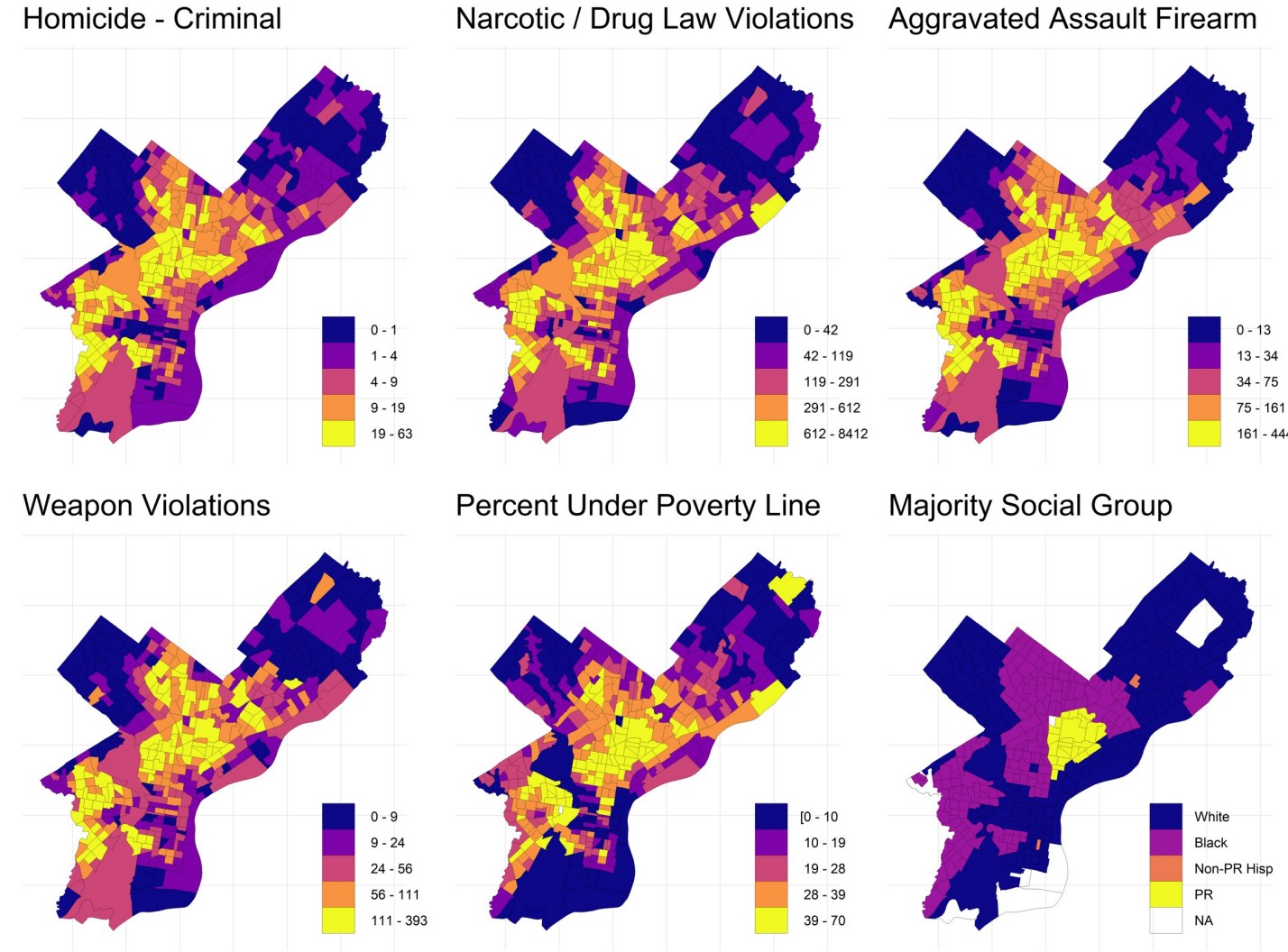

**Fig 2. Census tract level counts of crime-related incidents.** Based on data from the Philadelphia police department, as well as the percent of individuals living under the poverty line and the majority social group from 2012 to 2016 American Community Survey data.

Tito with whom he was very close, had achieved meteoric success in the narcotics economy. Leo had seen these men earn many thousands of dollars per week, as well as command the respect and admiration of their peers, by taking over boss-level roles in the neighborhood business of peddling narcotics.

In this context, Leo made the decision to enter the narcotics economy, like so many other young men in the area before him. He started out at the lowest ranks of "hustler" making hand-to-hand retail sales. His employer was Raffy, the neighborhood *bichote*, who "owned" the block. *Bichote*—a term for "drug boss"—is a Spanglish double entendre of the phrase "big shot," and Puerto Rican slang for a large phallus. Whenever *bichotes* arrive on a block they become the central organizing point of power, influence, cash flow, risk and above all employment.

To stay alive, maintain control of their territory, and remain free from incarceration, *bichotes* must walk a tightrope between respect and fear. Raffy's survival was predicated on his

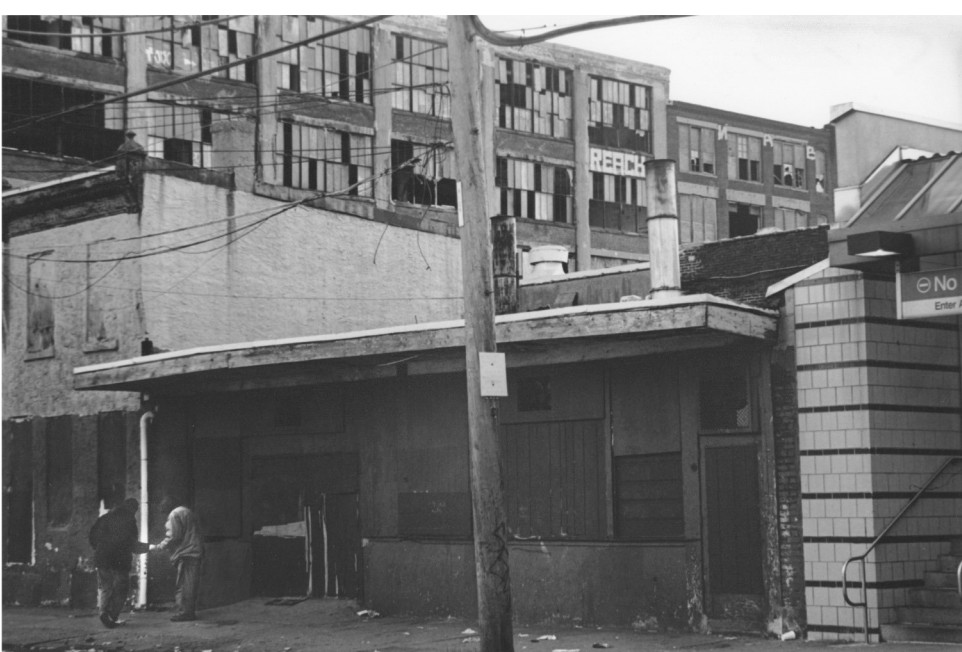

**Fig 3. A narcotics transaction in the ethnographic field site in Puerto Rican inner-city Philadelphia.** Hit hard by deindustrialization and disinvestment from the inner-city, young men born in this neighborhood often find themselves working in the lowest-level of the retail narcotics economy, selling heroin and cocaine in the shadow of the factories that used to employ their new immigrant grandparents. Photo by George Karandinos.

ability to command fear, so as to prevent dozens of other ambitious young men from taking over his corner by force. Fear, however, is not enough to protect a bichote from being reported to the police in the long-term. He had to earn the respect of the neighbors by cultivating a reputation for generosity and a charismatic ability to keep the neighborhood safe from criminals and interlopers. As we have described in other publications, a *bichote* who is seen as overly greedy or too permissive of interpersonal violence or aggressive behavior towards women on the block could expect to be ousted by neighbors repeatedly calling the police with precise information about the location of his caches of drugs, guns and money [28]. For *bichotes* who can navigate these dynamics—keeping potential usurpers at bay and the neighborhood residents happy, or at least begrudgingly tolerant–the payout was enormous. During a single twelve-hour sales shift, ten to fifteen thousand dollars of cash in ten-dollar bills would routinely change hands on a single drug corner in this desperately poor inner-city neighborhood. We saw drug bosses rapidly accumulate hundreds of thousands in tax-free cash income, buy multiple fancy cars and invest in local rental properties. We saw the "American Dream" of many others, however, crash and burn in a fusillade of bullets or in court fees, civil assets forfeiture cases and/or behind bars.

*Bichotes* run their neighborhood narcotics business much like other retail endeavors, with a hierarchy of customer-facing and administrative roles. They hire managers, called "caseworkers", to run 8- or 12-hour sales shifts at a specific "*punto*" (sales point), and "runners" who move drug supplies and cash between the sales points and safe houses. Caseworkers are responsible for hiring and managing "*joseadores*" (Spanglish for "hustlers") who make the hand-to-hand sales to clients of packets of heroin and cocaine in $5- and $10-dollar ink-stamped and scotch tape sealed packets. There is highly differential risk at each level of this retail narcotics hierarchy that reflects risk-taking and responsibility. The hustlers are the most visible and along with the clients, they face the highest risk of arrest and chronic incarceration.

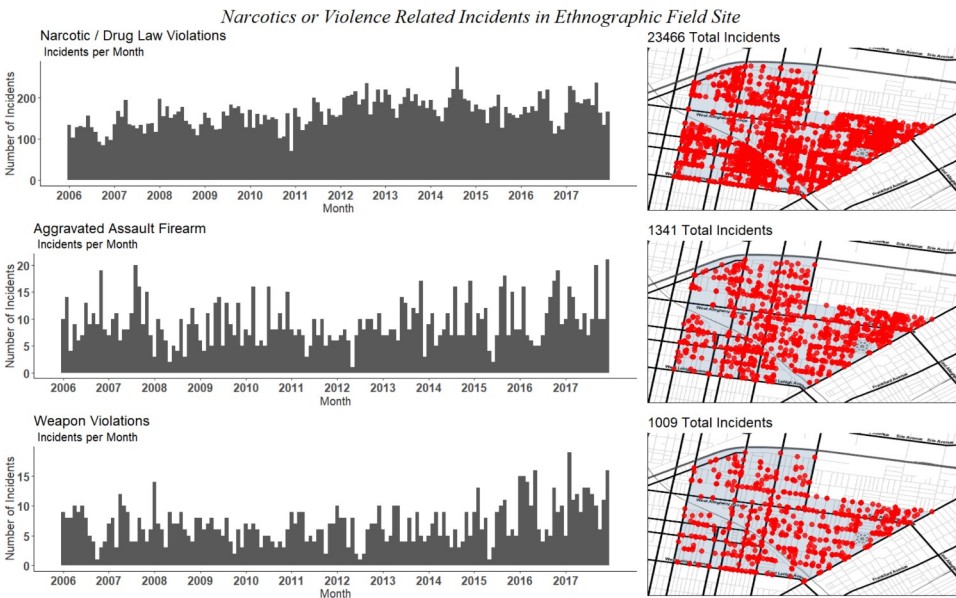

**Fig 4. Crime-related incidents in the ethnographic field site from 2006–2017 by type of incident.** Shown by monthly incidence (left) and spatial distribution (right).

There is a near constant police presence in the neighborhood, and drug busts are extremely common. Yet the police often fail to conduct the more complex surveillance work needed to identify caseworkers, runners and bichotes. Instead they arrest whichever customers and hustlers happen to be present at the moment of their raid. Fig 4 highlights how each month there are approximately 200 drug arrests—about 7 per day—in our relatively small field site, to say nothing of the many raids that do not result in arrests.

Each higher level of management in the narcotics hierarchy earns a greater portion of the profits and is also increasingly insulated from the ever-present risk of arrest. *Bichotes* ultimately earn a lion's share of the profit and most spend minimal time at the sales points, lest they attract too much police attention. In this way, the neighborhood sales point represents a self-contained instance of exploitative capitalism—the lowest-level employees experience the most risk while the business owners accumulate windfall profits, all situated in the poorest and most infrastructurally devastated corner of inner-city Philadelphia.

In tenth grade, Leo started out working at the lowest levels of the narcotics economy, selling packets of heroin and cocaine and dodging police raids, for several years. His big break came, ominously, when his older brother Tito was arrested for the accidental killing of his business partner, after having lasted only lasted 3 months as a bichote. Tito rose to the position by skillfully negotiating the rights to a corner from the wife of its previous owner—who had suddenly been shot dead by the brother of his caseworker whom he had failed to bail out of jail. The wife of the deceased *bichote* trusted Tito who had been a schoolmate of hers and a runner for her late husband. She offered to let him run the block for the exceptionally low rent of $500 per week (the block we lived on often rented for $5000 per week) because she was scared her husband's cousins might steal it from her. Tito happily agreed and established a partnership with a friend who owned a .357 Magnum pistol—to guard against retaliation by the murdered *bichote's* cousins. After three months of successful earnings, Tito accidently shot his partner dead while posing for selfies in the midst of a drunken celebration. Tito was charged with homicide and faced a 17- to 34-year sentence. As a result, Tito's younger brother Leo inherited the *bichote* title on the block. He eagerly rose to the challenge, despite the inauspicious

circumstances of his promotion, and he was determined to "do it right" and delegate all the risk to his caseworker and hustlers.

## The role of gun violence in the maintenance of order in the narcotics economy

Tito's choice of business partner—a young man who owned an imposing automatic handgun—is indicative of the importance for a *bichote* of maintaining the perception of a capacity for violence. In our field site the sale of cocaine and heroin represents an extremely lucrative market located in a deeply impoverished area. In a single day, hundreds of thousands of dollars of untraceable cash routinely flow through a neighborhood where the majority of residents live well below the poverty line. In this context, Tito, and Leo after him, needed to be able to perform violence at a moment's notice to protect their reputations and maintain monopoly control of profits. In a clandestine market there is no legal recourse should product or profit be stolen by an employee or a competitor. Order, therefore, generally is maintained with violence or the threat of violence, both within and between *bichote*-led retail businesses. Ultimately it was this need to maintain control that tragically resulted in the near-loss of life of Leo's employee, and caused Leo to lose 10 years of his life behind bars with no access to drug treatment or vocational, educational or mental health services.

As a precocious 18-year-old teenager who had stepped up to fill his brother's shoes as a *bichote*, Leo had been routinely bullied, threatened, and disrespected by his jealous and slightly older employees whom he had hired to work for him. We visited Leo in jail where he anxiously reflected on why he had pulled the trigger:

*Oh man, I got into some dumb shit. Real stupid! It was all over some nut-ass shit. I had this young bol [man], Adrian, out there hustling for me and I went around the corner to advertise my stamp [shout out his heroin's brand name to passersby]. When I go back, the work [cache of drugs] ain't there, so I'm like, "Adrian, damn, you're the only person sittin' here, like, what's up? Where the work go?"*

*[Imitating ostentatious innocence] "Oh, I didn't touch nothin'" . . . this-an' that. Then he wanted to get all hype, so he called his peoples—all of his cousins. So I go back to my crib and I grab the strap [gun] and I come back. [Head in his hands his voice cracking] I don't know, everything was just moving so fast, like. I ain't really know what to do. I was gonna smack the shit out of him with the jawn [indefinite pronoun, in this case meaning the handle of his gun]. But he kept talking. I raised my hand at him but he dipped back. And all his peoples was standin' there, I was thinkin' in my head, like [setting his face into a threatening frown], "Damn, if one of his peoples got a gun . . ." And Adrian like [taunting voice]. "You a nut-ass nigga! You ain't gonna be treating me like a nut' . . . This-an'-that . . ."*

*I'm like, "What!" And I pulled the jawn [gun] out. But he was just like, "Nigga you not gonna do shit." And he came at me. So I shot him, but just once so he could get away from me. That the first time I ever shot somebody. And I thought I was gonna be like hesitant. But I didn't even hesitate. It was just like a spur of the moment thing.*

*Afterwards, from my crib I had called one of his peoples. He told me they found the dope and I told him, "Look, when Adrian get better, we could rumble [fist fight]." But they told me Adrian was like almost dying in the hospital 'cause the bullet almost hit his main artery. I'm thinking in the back of my head, "Damn, I didn't want all that to happen . . . I just did some dumb shit." Next thing I know, the police come running up in my crib. "Where the gun at?" And started rippin' the house apart.*

All Leo had wanted to do was threaten his employee into returning his $500 stash of heroin, but the desire to maintain the appearance of control over his business, and the very realistic fear that any of his employee's family members might also have a gun, led him to use lethal force instead of a simple fist fight or pistol whipping as he had intended. This was just one of many fatalities or near-fatalities that we documented as rival individuals vied for control of the lucrative sales points on almost all the blocks surrounding us. We documented nearly a dozen *bichote* transitions during our time at the field site. These shifts in local power were often precipitated by the killing of a *bichote* for a perceived offense, or his or her incarceration. Each time bichote stability was upended in this manner, a violent scramble for power ensued, often among multiple aspiring rivals.

This central role of violence in the narcotics economy has resulted in the exposure of local residents to a high prevalence of violent events. Fig 4 shows the spatiotemporal pattern of crime-related incidents according to the Philadelphia police department. From 2006 to 2017, in our small, ten square block field site there was a monthly average of 9 aggravated assaults with a firearm, and 7 weapons violations. It should be noted that these likely represent only a fraction of the true burden of violence in this community, as many incidents go unreported.

In addition to the intentional and functional violence that surrounds power struggles for the narcotics market in inner-city Philadelphia, the prevalence of narcotics traffic, guns, and cycles of violence led to a number of unintentional deaths or close calls for residents of the area who had nothing to do with narcotics. For example, in one instance of retaliation-gone-wrong, numerous shots were fired, in the middle of the day, through the front door and into the living room of a peaceful family of undocumented Central American migrants. Luckily no one was harmed because the whole family was out working at legal jobs in downtown restaurants as dishwashers. Their door had been mistaken for that of their next-door neighbor, a young man who had recently accepted a plea deal that was interpreted as proof that he had "snitched" on his *bichote* employer. Though unharmed, the family packed up and left the neighborhood the next day, presumably terrified of future violence or deportation stemming from the incident. During our time in the field site we witnessed many such instances of accidental violence affecting residents who were not directly involved in the narcotics trade, and not all ended as favorably. In one incident that was frequently recalled on our block, one of our neighbor's high school-aged sons was killed in crossfire during a mid-day shootout stemming from a drug dispute. He had been frantically trying to usher the children playing on the block out of harm's way when a stray bullet caught him in his neck.

Violence is also central to the policing strategy employed by the Philadelphia police department. During our fieldwork Philadelphia newspaper reports—including one Pulitzer Prize winning investigation—documented several dozen police brutality and corruption scandals [40,41]. Early in our fieldwork, a member of our ethnographic team recording interviews with sellers (PB) was knocked to the ground during a routine police raid, handcuffed facedown on the pavement, and then kicked with such force that several of his ribs were cracked. He was then arrested on false narcotics possession charges. Whites present in the neighborhood were profiled by the police as addicted customers and the white members of the ethnographic team were regularly harassed, detained, and insulted by the police as they walked home to the block from the subway.

More commonly, the victims of police brutality were the Puerto Rican neighborhood residents. The following fieldnote describes George's observations of an instance in which police—responding to a call about a drug-related shoot-out—brutally assaulted a young Puerto Rican male bystander, while failing to apprehend anyone involved in the shooting:

*Sitting on the stoop I notice an immobile young man lying limply at the feet of the police with his hands cuffed behind his back. They ignore him while they search the vicinity. A police*

*flashlight illuminates the blood-stained concrete near the youth, and I see that he is lying with his neck at an odd angle; his face totally covered in blood. He doesn't move for several minutes, and I wonder if he is conscious.*

*One of my neighbors tells me that there has just been a shootout which I had mistaken for fireworks: "Four black guys got out and went to their trunk, pulled out guns and cocked them and then walked down the block. I went inside, and then I heard the shots." Another neighbor adds that the young man lying on the concrete had run from the police, even though he didn't have anything to do with the shoot-out. "He tripped and dove halfway under a car. The police caught him, and kicked him in the face, hard, pulled him out, and beat him to a pulp right on the street."*

*Someone suggests that he might've had "work" [drugs] on him, prompting him to flee, even though "the cops weren't gonna do shit to him if he had just stood around, but he panicked." I suggest that they probably didn't find anything on him since they were looking for so long. Someone else informs us that he only ran because he was on probation and was afraid they would think he had something to do with the shooting.*

*Instead of calling an ambulance, the officers roughly bundle the handcuffed young man into the back of a police car, half-carrying him since he can barely walk. They slam the door shut without securing him to his seat and without saying a thing to us, they speed off leaving a puddle of coagulating blood on the street and the stench of burnt rubber from their tires. Everyone standing around agrees it was "fucked up" that the cops beat someone so savagely "when he hadn't done anything." Someone suggests we should call a lawyer, and another suggests calling "the supervisor" of the officers to report an incident of brutality. "Fuck that," someone says dismissively, "there was a white shirt [captain] out here when it happened. No point in complaining to the perpetrators!" A couple of kids come by to take pictures of the pool of blood, furious that their friend had been brutalized so badly by the police. I just watch in shock.*

We witnessed several other such instances of egregious violence directed towards neighborhood individuals by police. These incidents alienated residents from the police, adding yet another layer of victimization to a community already plagued by violence from within [28,32]. Importantly, violence perpetrated by police against neighborhood residents is unlikely to be represented in the statistics we present here, or in public health databases generally [42].

When young men involved in the narcotics economy eventually found themselves incarcerated, the performance of violence again emerged as vitally necessary for survival in the overcrowded and terrifying world of the Philadelphia County Jail. For example, Leo's older brother Tito who was arrested after accidently killing his business partner, was brought into the jail visiting room when we came to visit:

*This unit is crazy man. A lot of people don't know what's going on yet with their case. They stressin'. They have that uncertainty. They don't know if they are going to go home soon, or if they aren't ever goin' home. Plus, we in close custody. They got us on lockdown half the time because of some shanking [stabbing]. There aint' shit to do. You just sit in your cell all day bored and frustrated. That's half the reason there so many problems. We might kill each other over 10 minutes on the phone. Or hot water in the shower, or whatever.*

*Out in the street I knew how to resolve a situation, you could talk to someone out there and maybe it didn't have to come to any violence. In here there is no choice. You can't just let them treat you like a bitch 'cause then everyone be sayin', 'He a pussy. He ain't gonna do anything.' And walk up in your cell, "Look nigga gimme all that, or I'm'a fuck you up." I done seen it too*

*many times man. No one is going to talk about me like that. All I have in here . . . [choking back tears] is my pride. I'm not letting nobody take that away from me!*

*I just got in a fight with some black bol and look, the motherfucker bit me! We had words earlier at the phones, and he kept runnin' his mouth. But I let it go. I wanted to be peaceful, you know, I have a lot on my mind. I have to go to court tomorrow. But nigga came into my cell and [making a punching motion] snuck me in the back of the head. Then stood there lookin' at me like I wasn't gonna' do nothin'. Like I'm a pussy.*

*I guess 'cause I'm small and I'm Puerto Rican, and I came in here quiet, minding my business, people think they can fuck with you. That's what I get for trying to keep to myself. I know if I came in here like a savage then he wouldn't done that.*

It was a matter of common sense that cultivating a reputation for violence was essential to avoid being victimized while incarcerated. Young men entering prison would often aggressively attack anyone who seemed to vaguely threaten or disrespect them, in order to avoid being targeted for subsequent rape or murder. This mechanism of self-protection also had the unintended effect of trapping incarcerated men into ever-extending prison sentences with add-on charges for violence committed while incarcerated, as well as trauma-inducing punitive isolation lockdowns that further increased their desperation and rage. Carceral experiences of violence and trauma further cemented the propensity for violence as a core element of a man's identity. It increased the chances that once released he would respond to challenges with hyper-aggressive displays of violence that ultimately feed back into ongoing cycles of mortality and morbidity in the neighborhood.

In the midst of power structures that privilege the perpetration of violence for economic and social success and protection, the easy access to automatic weapons raised the stakes of each instance of violence and increased the total mortality levels. During our jail visits, Leo reflected on how he had obtained the gun that he had used to shoot his employee, and the role of easily accessible guns in his life:

*I bought the jawn [weapon] off one of my homies. It was a big-ass chrome forty [.40 mm]. I put $300 and my bol Freddo put $300. We was sharin' it. It was real cheap 'cause somebody probably already done did something with it.*

*I'm a gun freak, I love them too much. They just come to me. Like, [Imitating a sales pitch] 'Yo, I got a shotgun $100. Real cheap! . . .' [Voice filling with energy] 'a nine [9mm] . . . a forty . . .' And, I'm like, '[Eyes lighting up] I need that!' I don't know why. I got to leave them alone . . . I had so many guns in the house, I'm surprised that my mom didn't just get rid of me [tears welling up and putting his head in his hands].*

In a cultural and economic context that places substantial emphasis on the perception of a capacity for violence, and with ineffective gun control laws in place, it is unsurprising that gun fetishism emerges around increasingly powerful automatic and semiautomatic weapons. As we saw with the story of Leo's near-killing of his employee, the ubiquity of powerful firearms and fear of retaliation encourages the pre-emptive use of lethal force, even when lesser measures could suffice: "I was thinkin' in my head, like, 'Damn! If one of his peoples got a gun. . .'

## The production of violence as a structural and cultural phenomenon

The economic, structural drivers of violence are of paramount importance in understanding the bloodshed we witnessed in our field site. Poverty and lack of formal employment ultimately

play a central role in the production of the narcotics economy, as well as the violence that accompanies it in the context of prohibition. Nevertheless, this macro level line of reasoning does not account for the greatly elevated levels of narcotics traffic in low-income majority-Puerto Rican areas relative to majority-black neighborhoods with similar levels of poverty. Based on our observations, both highly-impoverished black and Puerto Rican neighborhoods had narcotics retail businesses operating on their corners. Nevertheless, the scope of sales at black areas paled in comparison in terms of volume and organization to their Puerto Rican counterparts (S1 Fig). The following fieldnote illustrates the pace of open-air sales witnessed each time we walked to our apartment from the nearest subway stop [24]:

*Before I have walked halfway down the subway platform stairs I am hailed with, "Works [syringes]! Works! Sub [Suboxone pills], sub, sub!" As I step onto the sidewalk an emaciated white injector offers to take me to a corner "that's poppin' today." He assures me that he was given a sample less than an hour ago ". . . it's a 10 [highest quality rating]." I have learned to shake my head, mumbling, "I'm good," and continue rapidly down the sidewalk. I find myself in the midst of a stream of mostly white injectors in various states of emaciation and ill-health. They are fanning out from the subway entrance. hurrying through the labyrinth of surrounding narrow one-wayside streets*

*A twenty-something-year-old young white man in a Penn State sweatshirt with his baseball cap tilted backwards is walking just a little too fast and too eagerly next to me. He could look like he just walked off a college campus but is 20- or 30-pounds under-weight. He raises two fingers of his right hand in what I mistake to be a victory sign and peels off across the street towards a Puerto Rican teenager who is crouching by the tire of an SUV and pulls out two packets of heroin for him from underneath the chassis. They make a quick one-handed exchange. Spinning around, he thrusts his hand down the back of his pants, stashing the heroin in his rear before heading straight back to the subway.*

*Ahead of me there are two couples, both consisting of a young, skinny, scantily dressed woman walking more confidently than their older boyfriends. But most of the injectors around me who are on their way to buy heroin are single men walking alone or in duos, sometimes trios, in temporary nervous alliances for protection and information about "what's best today." Others are scanning about looking for an acquaintance to guide them to "the best dope" in return for a tip or a taste. A burly, white middle-aged man in paint-splattered pants presumably taking a user's break from a contractor's job, or else still buff from weightlifting during a recent bout of incarceration, asks me, "Is Godfather open today? Have you tried it?" I shrug my shoulders and look away, but another younger more emaciated white 30 something-year-old man with a big friendly smile, overhears the question and shuffles over, his foot wrapped in a filthy bandage, "I had some. Godfather's poppin' today." In the same breath, the painter anxiously snaps back, "How long ago?"*

*The flow of addicts, many of whom look like the walking wounded, has now reached the next corner and we are greeted by two physically-fit, clear-eyed Puerto Rican teenagers dressed in the latest hip-hop style, shouting "DOA [brand name] DOA!" and "powder [cocaine], powder, powder, powder. . . What you need?" followed by a fainter chorus of "works, works, works" coming from a set of older, broken-down-looking whites who are standing almost deferentially further away against an abandoned rowhome. They are clearly subordinated to the younger Puerto Rican heroin and cocaine street sellers. These choruses repeat themselves half a dozen more times on just about every block, sometimes again halfway through the block through which I walk until I reach our apartment. On our block, the brand name has been "Dead End" for the past three months.*

Most corners in the 300+ square blocks surrounding us had a near constant stream of customers moving through the area, mostly white outsiders. In comparison, the African American sections of North Philadelphia appeared to have a much lower volume of customers, and most were African American residents of the same neighborhood. Given that there were both black and Puerto Rican areas with similarly high levels of poverty and chronic unemployment (Fig 2), a simple economic argument linking poverty to narcotics sales is insufficient to explain the exceptional level of sales to white outsiders in poor Puerto Rican neighborhoods.

As we came to understand, the concentration of the retail narcotics market in our field site needs to be understood in the context of a deeply racially divided city with long standing tensions between black and white communities. At the time of our fieldwork, the vast majority of customers seeking heroin in Philadelphia were white, reflecting the opioid epidemic of the time that was mostly concentrated in low-income white populations [43,44]. These white customers were much less likely to enter black neighborhoods to buy heroin for several reasons. Phenotypically they stood out much more in a racially-segregated black area and they were easily and immediately profiled as addicted customers by police, exposing them to arrest or harassment. Black-white animosities also increased the possibility of being heckled or mugged when seeking drugs in black neighborhoods. As one African-American dealer put it to us, while we stood observing the frenzied slew of customers flowing through a Puerto Rican block:

> It don't work this way in South Philly [a predominantly African-American Area]. Everybody be out for they self. Papis [local racist term for Puerto Ricans] are smart, they get that white money. All of it! And the white junkies keep coming back with more! If these people walked onto my block with cash in their hands, someone would take their money. Matter of fact [laughing] I might.

This contrasted sharply to the attitudes of Puerto Rican residents, who tended to view the emaciated white clients with some sympathy and even occasionally with admiration for their light complexions. Philadelphia has a long and fraught history of racial segregation and race riots. In a particularly iconic example that is still a political flash point in the city, a mere 20 years before we began our fieldwork, the virtually all-white Philadelphia police force fire-bombed an all-black residential neighborhood in the name of fighting the black power movement, destroying 61 homes and killing 11 people [45,46]. In this context of extreme racial tension, Puerto Rican neighborhoods have served as a neutral meeting ground where white clients could more comfortably navigate through the inner-city to procure heroin.

Our ethnographically-derived understanding of the hyperbolic nature of drug sales in our field site can be confirmed by looking at police records showing an extreme concentration of arrests for narcotics in impoverished majority Puerto Rican areas. As Fig 5 indicates, across the board higher levels of poverty are correlated with higher levels of narcotics arrests, yet this gradient is much steeper for Puerto Rican areas relative to their black or white counterparts. As is clearly indicated by Fig 5, the most impoverished Puerto Rican neighborhoods have hyperbolic levels of drug arrests compared to the poorest black or white areas. The fitted lines on Fig 5 also show the predicted results of a statistical analysis of the poverty-narcotics gradient, separately by majority social group. In Fig 6, we predicted the expected rate of narcotics incidents for a poverty rate of 60%, similar to our field site (Fig 6), and compare the ratios of these predictions for each racial group. This allows us to control for poverty and show the association between race and levels of violence- and narcotics-related crime. The predicted values of narcotics-related crime for highly-impoverished, majority-Puerto Rican neighborhoods were significantly elevated compared to that of majority-black and white areas, with ratios of

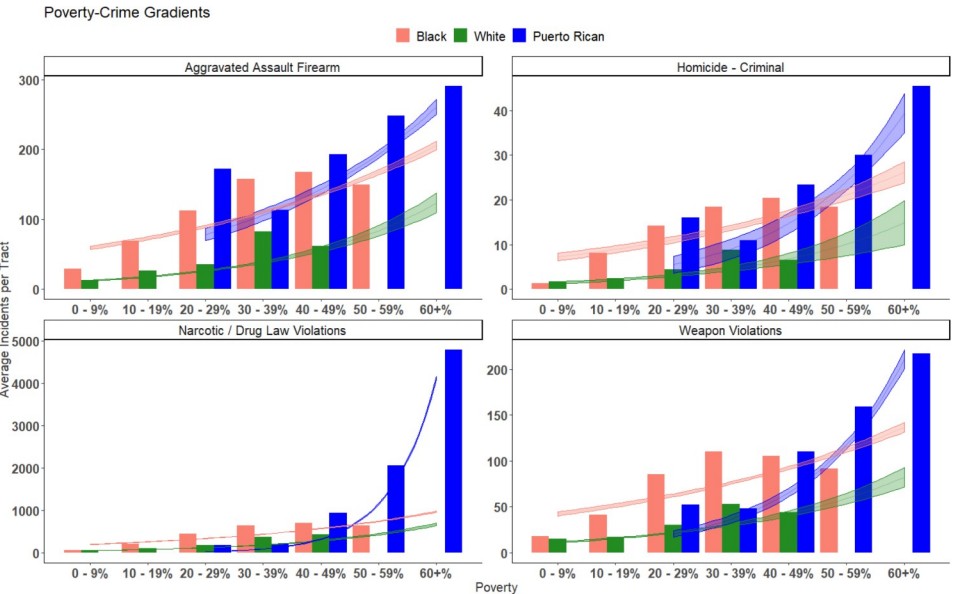

**Fig 5. Poverty-crime gradients by racial group.** Bars represent the average number of crime-related incidents per census tract, from 2006 to 2017, by percent of the population living in poverty, separate for majority-Puerto Rican, -black, and-white areas. The fitted lines represent model predictions from the Poisson regression, with 95% confidence intervals.

4.2 (95% CI: 4.1–4.3) and 6.1 (5.8–6.4) respectively (Fig 6). This indicates that at a poverty level of 60%, majority Puerto Rican areas were predicted to have over 6 times more narcotics crime relative to majority white areas. The relationship between poverty and violence has also been well documented. It has become relatively commonsense that in the United States, higher poverty rates are correlated with a greater burden of interpersonal violence. Based on our observations that violence in our field site stems directly from the narcotics economy, we hypothesized that the poverty-violence relationship would be steeper for the most-impoverished Puerto Rican areas. Even controlling for the deeply entrenched poverty, we would expect that they have excessive levels of violence due to their key position in housing the retail narcotics markets of the city.

A quantitative examination of police crime data suggests that this hypothesis has merit for several categories of violence. As shown in Fig 5, there is a clear poverty-violence gradient seen for all social groups. Nevertheless, it is clear that the most impoverished Puerto Rican areas are the most burdened by violence of any part of the city, when considering homicides, assaults with a firearm, or weapons violations. Nevertheless, it is important to consider that our field site also represents the most impoverished corner of Philadelphia, and therefore we used a statistical analysis to assess if the level of violence exceeds what would be expected purely on the level of poverty alone. That is to say, that the poverty-violence gradient is steeper at the most low-income end of the spectrum for majority Puerto Rican areas relative to majority black or white areas, and the observed differences are not due to chance alone.

Poisson regression results confirm our descriptive observations of the quantitative data. Poverty is significantly associated with homicides, assaults with a firearm, and weapons violations, for all groups. Furthermore, Fig 6 indicates that at a 60% poverty level, majority-Puerto Rican status is associated with significantly elevated levels of violence, for each category of violent crime examined. At lower levels of poverty sharp disparities can be observed for majority-black neighborhoods, which often have several times the level of violence compared to their

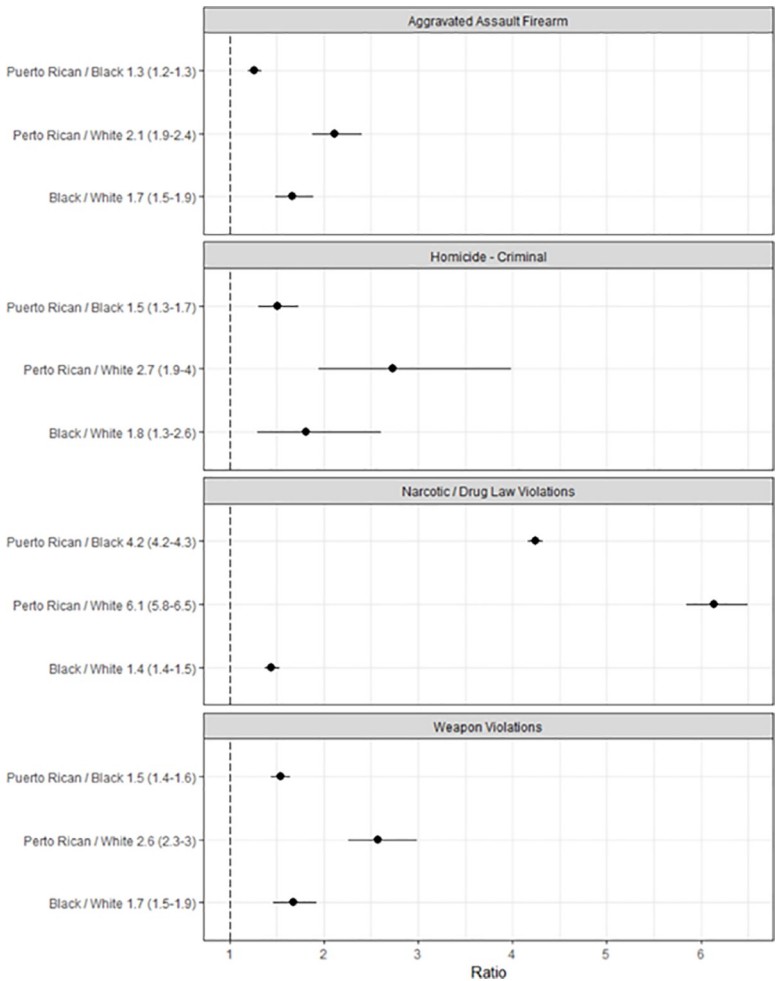

**Fig 6. Racial disparity among maximally impoverished neighborhoods.** The ratio, with 95% confidence interval, between social groups of predicted levels of violence- or narcotics-related crime from 2006 to 2017, for a poverty level of 60%, similar to our field site in the most impoverished corner of inner-city, Puerto Rican Philadelphia. For example, a ratio of 6.1 between whites and Puerto Ricans for narcotics related crime indicates that majority Puerto Rican areas were predicted to have over 6 times more narcotics crime relative to majority white areas, at a poverty level of 60%.

equally impoverished majority-white counterparts. Nevertheless, the highly impoverished majority Puerto Rican areas clearly represent the most burdened part of the city in terms of poverty and also violence- and narcotics-related crime.

These trends are also reflected in the per-capita murder rates observed across the 2006 to 2017 period. The city-wide murder rate was 20.0 per 100,000 persons per year, compared to 5.8 for majority-white areas, 30.8 for majority-black areas, 39.0 for majority-Puerto Rican areas, and 62.0 for the core narcotics area that was our field site, identified in Fig 1.

## Conclusions

### Documenting and humanizing inner-city violence with mixed methods research

In this mixed methods study, we document racial and economic disparities in exposure to violence in inner-city Philadelphia. The maps contained in this study paint a troubling, yet all-

too-common picture, of a hyper-segregated city in which poverty and violence levels correlate strongly with race/ethnicity [16,24,24]. Our especially-poor inner-city field site had a per capita murder rate over ten times higher than the average for majority-white areas, and three times the average for the city.

Clearly, economic factors play a key role in driving the violence we witnessed. Young men such as Leo and Tito have little opportunity for social mobility outside of the narcotics economy, and role models for other paths are essentially non-existent. The extreme poverty in Puerto Rican inner-city Philadelphia stems from its rapid deindustrialization, the history of colonial domination of the island of Puerto Rico, and the increasing disinvestment of the public and private sectors in inner cities [4,18,28]. This poverty has only been exacerbated by recent cuts to welfare and social services that render subsistence survival at the lowest economic rungs even more difficult [39]. The narcotics economy has arisen to fill the void, providing employment and a chance at social mobility, but at a steep cost in risk for violence and incarceration.

The logic of prohibition for a highly-sought-after product such as heroin or cocaine today (or alcohol in its time) inevitably creates a lucrative black market that is protected by violence, and systematically inflates levels of crime in society [47]. In the punitive political context of the War on Drugs, and the hyper-incarceration of non-white individuals involved in the narcotics economy, young men in our neighborhood had little chance to escape incarceration, injury, and/or, at the extreme, death by homicide [48]. In this way, powerful economic and social structures directed young men of color through a series of decisions and punitive institutions involving a high risk of victimization and/or perpetration of lethal violence.

Clearly, the bloodshed we observed as a daily phenomenon in our field site must be understood as "structural violence" [17]. Our friends were rendered "structurally vulnerable" to violence from the social and economic forces that create average levels of violence many dozens of times higher in poor non-white neighborhoods compared to affluent white areas just a few miles away [16]. These "social determinants" play an outsized role in shaping the economic and health outcomes available to residents of our field site [20,49]. Nevertheless, through contextualizing these disparities in a mixed-methods assessment of both structural and cultural factors, we see that to understand patterns of violence we need to employ an intersectional analysis involving not only economic drivers but also racial dynamics, spatial segregation patterns, and the individual-level experiences of these forces [50].

A simplistic structural assessment of poverty is insufficient to explain the group-specific vulnerabilities to violence we documented in Philadelphia. Even controlling for poverty, the homicide rate in our field site was nearly three times higher than a hypothetical similarly-impoverished white area (which, it is important to note, does not actually exist). The exceptional vitality of the retail narcotics trade and the violence it spawns in Philadelphia's majority-Puerto Rican neighborhoods reflect the unique social role of these interstitial areas sandwiched between poor white and black neighborhoods that converts them into a neutral racial meeting ground in hyper-segregated Philadelphia. The predominantly white, indigent customer base is much more likely to seek heroin and cocaine in phenotypically more diverse Puerto Rican neighborhoods, rather than black neighborhoods where they are more conspicuous to law enforcement, and more likely to be assaulted in the context of deep racial animosities. Gun violence consequently cannot be understood simplistically as the direct result of poverty and racism alone. Instead, the nuances of the particular racial and economic history at hand must be appreciated to understand, and subsequently ameliorate racial/ethnic disparities in exposure to violence. For this reason, it is paramount that firearm violence prevention programs take both a cultural and structural approach to ameliorating the epidemic of gun violence faced in the United States.

We provide a nuanced analysis of the particular vulnerability of Puerto Rican communities in Philadelphia, which has not been thoroughly documented in the epidemiological literature. This represents just one small corner of America's epidemic of firearm and interpersonal violence, yet it has implications for socially-patterned violence in inner-city areas throughout the country. There are other pockets of culturally patterned violence with very different mechanisms, such as firearm suicide among low-income white males in the Southern US and especially in rural communities [51]. This highlights the need for further contextualization of epidemiological data describing gun violence with close examination of the particular cultural and structural mechanisms that are relevant for each hot-spot of bloodshed.

## Limitations

As all quantitative studies, this analysis is limited by its data source. The use of police data to study violence in our inner-city field site could be seen as ironic, given the disruptive and often violent nature of the police interactions we witnessed on a regular basis and the racially selective enforcement of the War on Drugs. Yet it is most indicative of the unfortunate reality that the only public institution that reliably interfaces with this community, and has granular data related to its victimization, is the Philadelphia Police Department. Our results must therefore be considered in the context of the limitations of these data. It is certain that police data undercount the prevalence of both violence and narcotics, and potentially in irregular ways that could bias the results. Nevertheless, the large magnitude of the disparities presented here make it unlikely that any bias in the data could reverse them, especially given their consistency with our ethnographic observations. Nevertheless, it is possible that a bias in the police force against Puerto Rican neighborhoods could lead to greater police attention in those areas, subsequently overrepresenting violence or drug crime in those areas. Furthermore, the white/non-white disparities observed here may be magnified by racial biases held by police officers, wherein drug trafficking in white areas receives less attention that its equivalent in non-white areas.

It should also be noted that our quantitative analysis is ecological, we are describing neighborhood level associations at the census tract level. There are minority individuals in each tract who do not pertain to the majority social group whose identity defines the major dynamics of the neighborhood. Nevertheless, it is these neighborhood-level effects we are most interested in capturing in this study, and the extreme degree of racial segregation in Philadelphia limits the complications of relying on neighborhood-level variables to describe social groups. We also use pooled data representing 2012 through 2016 for sociodemographic indices, which represents a moving average that does not reflect any single point in time during this window. For limitations related to ethnographic data collection we will refer to more extensive, previously published discussions of our ethnographic methodology [24,28,29,32]. Our work is most representative of the time period during which the intensive fieldwork was conducted (2007 through 2013), although our periodic return visits to the field site indicate that the dynamics described here remain largely predominant.

## Implementing structural solutions to structural issues

Gun control represents the most obvious key strategy to control the generalized crisis of firearm violence throughout the United States. Easy access to discounted "used" automatic and semi-automatic guns raises the already high stakes of struggles for power in the crowded and profitable inner-city narcotics market. In many instances we saw accidental and intentional fatalities in our field site that would likely not have occurred if the bar for attaining a weapon in the United States were substantially higher. Nevertheless, it is difficult to imagine how moderate gun control measures could make a meaningful impact on the pattern of violence we

observed in our field site in Philadelphia. The inner-city is already awash in legally and illegally obtained firearms, and it is difficult to conceive of a politically-viable intervention that could rapidly empty the streets of these weapons.

The logic of participation in the narcotics economy, and the violence that surrounds it, cannot be overcome without structural interventions that address the deeper roots of American violence. Importantly, the policies of hyper-incarceration oriented around the War on Drugs have been profoundly counterproductive. They criminalize the residents of our field site while doing virtually nothing to assure their physical safety. Indeed, we observed that despite near-certainty of incarceration, young men continued to fill the lowest ranks of the retail narcotics market and readily served as foot soldiers defending the profits of their bosses, in hopes that others would someday do the same for them. Indeed, our observations suggest that hyper-incarceration actually fuels violence, as one of the most common destabilizing forces throwing a neighborhood into a bloody fight for control was the imprisonment of a *bichote*.

The election of Larry Krasner as Philadelphia's District Attorney in 2018 marked a dramatic retreat from the hyper-punitive approach characterizing the past half century of US criminal justice policy. Krasner ran on a platform of "ending mass incarceration." In just his first year in office, the city saw a 30% reduction in the jail population, as well as reduced parole and probation sentencing, and the elimination of cash bail for many non-violent crimes. At the time of this writing, Philadelphia was considering fully decriminalizing possession of all narcotics [52–54].

It can be argued that decriminalization possibly followed by legalization of narcotics would provide the only definitive solution to ending the highly profitable market that drives the astronomical rates of violence we observed in our field site. As long as prohibition exists, so will artificially-high profits that can only be defended with extra-legal violence. Similarly, the brutality, racial profiling and corruption of the police in the context of the War on Drugs alienate inner-city residents, pushing youth who have been victimized to pursue violent revenge rather than seek justice through the criminal justice system. Although the specifics of the casualties observed across the US and Latin America depend highly on the particularities of the social and economic circumstances, as long as narcotics represent a clandestine and profitable market it is likely that their presence will generate violence [47]. Beyond Philadelphia, prohibition has filled the streets of the United States and Latin America with powerful criminal organizations that terrorize populations and corrupt democracies in predatory pursuit of profits [47]. Various approaches to deconstructing this logic of prohibition have been proposed. In Europe, heroin prescription programs have had impressive success [29,55]. By offering individuals suffering from addiction a source of contaminant-free heroin, injected by a nurse in a safe facility, they have been able to dramatically reduce the social and health impacts of addiction—especially overdose. They also importantly remove customers and the profit motive from the narcotics economy, thereby representing a unique opportunity to decrease narcotics-associated violence. Instead of finding themselves forced into entry-level sales to support their habits, individuals are stabilized with less-damaging, contaminant-free heroin and rendered free to pursue other aspects of their lives. In this context, many find the stability they need to leave narcotics behind.

Outright legalization has also been enacted in limited settings—e.g. marijuana in the United States—and might represent a radical blow to organized criminal international syndicates. If narcotics were sold in safer forms by governments, without advertising, and coupled with accessible treatment and medical care, it would reduce the profit motive from the narcotics-driven violence that remains at crisis-levels all over the Americas. Nevertheless, as evidenced by the recent US opioid epidemic, widespread access to narcotics also has profound risks at the population level [43]. In either case, if the main source of economic survival is to be successfully taken from our inner-city field site, it must be replaced with other options for social

mobility. A Marshall Plan-level of formal job creation, expansion of educational opportunities, and infrastructure investment—bolstered by gun control, narcotics decriminalization/legalization, and rehabilitative criminal justice reform—represent the scale and type of violence interventions capable of driving structural long-term change.

> *There's old-ass people in here with white hairs. And them niggas ain't changed. You really gotta be strong to change. And I ain't gonna lie to you, I get sucked into doing dumb stuff.*
>
> *'Cause it's like a chain reaction. You come home [from prison] and you go back right to the same thing. This lifestyle is just so addictive—especially when you got a block [are a bichote]. You just wake up and you got money. You walk around the block and your workers passin' you some money. Next thing you know, "Yo, I'm done, come pick this money up."*
>
> *It's so easy. But it don't lead nowhere. Next thing you know you wind up killin' somebody 'cause he tried to kill you and you in this situation [shaking his shackles] ready to do more time. That's why I know I ain't gonna change if I come back to Philly. I just wanna leave Philly. I just really wanna get up and go. If I'm gonna go for the better, then good. If I'm gonna go for the worse, then it is what it is. I just wanna leave though. There ain't shit down here, man. Everything is nutty out here. It's like, it's impossible to make something, you know what I'm saying, if you wanna sell drugs you ain't gonna make it nowhere selling drugs. I rather take my chances somewhere else.*
>
> *So, I know if I leave I'm gonna go somewhere brand new. I ain't gonna know nobody. I ain't gonna sell drugs. I'll just go and be like, fuck it, let me get a job and go to school. I need to figure out a game plan to keep me away from the streets. I need to have a job before I get out of here. And I don't know how that's goin' to work. I ain't never had no job before.*
>
> > -Leo, at the outset of a ten-year prison sentence, reflecting on his hopes to escape the cycles of incarceration and violence inherent to his upbringing in the narcotics economy of inner-city Puerto Rican Philadelphia.

## Supporting information

**S1 Appendix. R code used for statistical analysis.**
(R)

**S1 Fig. Continuous map of crime and poverty in Philadelphia.** A version of Fig 2 in the main text that uses a continuous color scheme, instead of categorical, to highlight outliers. Maps show census tract level counts of crime-related incidents, based on data from the Philadelphia police department, as well as the percent of individuals living under the poverty line and the majority social group from 2012 to 2016 American Community Survey data. (DOCX)

## Author Contributions

**Conceptualization:** Joseph Friedman, George Karandinos, Laurie Kain Hart, Fernando Montero Castrillo, Nicholas Graetz, Philippe Bourgois.

**Data curation:** Joseph Friedman.

**Formal analysis:** Joseph Friedman, George Karandinos, Laurie Kain Hart, Fernando Montero Castrillo, Nicholas Graetz, Philippe Bourgois.

**Funding acquisition:** Philippe Bourgois.

**Investigation:** Joseph Friedman, George Karandinos, Laurie Kain Hart, Fernando Montero Castrillo, Nicholas Graetz, Philippe Bourgois.

**Methodology:** Joseph Friedman, George Karandinos, Nicholas Graetz, Philippe Bourgois.

**Project administration:** Philippe Bourgois.

**Software:** Joseph Friedman, Nicholas Graetz.

**Supervision:** Joseph Friedman, Laurie Kain Hart, Philippe Bourgois.

**Validation:** Joseph Friedman, Nicholas Graetz.

**Visualization:** Joseph Friedman, Nicholas Graetz, Philippe Bourgois.

**Writing – original draft:** Joseph Friedman, Philippe Bourgois.

**Writing – review & editing:** Joseph Friedman, George Karandinos, Laurie Kain Hart, Fernando Montero Castrillo, Nicholas Graetz, Philippe Bourgois.

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
