## [Decision Letter · Decision Letter 0]

20 Sep 2019

PONE-D-19-20254

The Unique Structural Vulnerability of Philadelphia’s Puerto Rican Inner-City to Narcotics-Driven Firearm Violence: An Ethnographic and Epidemiological Study

PLOS ONE

Dear Mr. Joseph Friedman, 

Thank you for submitting your manuscript to PLOS ONE. After careful consideration, we feel that it has merit but does not fully meet PLOS ONE’s publication criteria as it currently stands. Therefore, we invite you to submit a revised version of the manuscript that addresses the points raised by the two reviewers.

In particular, you need to discuss your theoretical framework of structural vulnerability and violence in the first section of the article. Also reconsider how you label your methodological approach. Is it mixed-methods or multi-methods? Justify your choice by referring to the relevant academic literature.

It is also strange that your ethnographic data yielded qualitative hypotheses about the racial/ethnic and social dynamics of firearm violence in Philadelphia. Rather than hypotheses, did you mean queries/questions that needed further investigation? 

The names Leo and Tito-- are these actual names of research participants or pseudo names used to protect their identities? 

Finally, how many interviews were conducted? How were the interviews analyzed? Where there any differences in views of interviewees in regard to your main research questions? 

We would appreciate receiving your revised manuscript by Nov 04 2019 11:59PM. To enhance the reproducibility of your results, we recommend that if applicable you deposit your laboratory protocols in protocols.io, where a protocol can be assigned its own identifier (DOI) such that it can be cited independently in the future. For instructions see: http://journals.plos.org/plosone/s/submission-guidelines#loc-laboratory-protocols

We look forward to receiving your revised manuscript.

Kind regards,

Cecilia Benoit

Academic Editor

PLOS ONE

Journal Requirements:

2. Please provide additional details regarding participant consent.

In the ethics statement in the Methods and online submission information, please ensure that you have specified (a) whether consent was informed and (b) what type you obtained (for instance, written or verbal, and if verbal, how it was documented and witnessed).

If your study included minors, state whether you obtained consent from parents or guardians.

If the need for consent was waived by the ethics committee, please include this information.

Reviewers' comments:

Reviewer's Responses to Questions

**Comments to the Author**

1. Is the manuscript technically sound, and do the data support the conclusions?

Reviewer #1: Yes

Reviewer #2: Yes

2. Has the statistical analysis been performed appropriately and rigorously? 

Reviewer #1: I Don't Know

Reviewer #2: Yes

3. Have the authors made all data underlying the findings in their manuscript fully available?

Reviewer #1: Yes

Reviewer #2: Yes

4. Is the manuscript presented in an intelligible fashion and written in standard English?

Reviewer #1: Yes

Reviewer #2: Yes

5. Review Comments to the Author

Reviewer #1: - This article presents an examination of the relationship between poverty, narcotics, and violence through a mixed-methods, ethnographic and epidemiological assessment, approach in Philadelphia. There are many benefits to this design, which provides rich insight into this issue. The ethnography itself is robust; however, there are some limitations with the quantitative data (discussed below). I cannot comment on the quantitative methods. The article is well-written and with minor revisions could be suitable for publication.

- The theoretical framework is not entirely clear, although it is touched upon in the discussion. The article needs to be situated within this framework early on.

- The article in well written and seem to be free of grammar issues. However, there are some terms or use of jargon, such an “integrated” ethnographic and epidemiological assessment, which are not entirely clear.

- I'm not convinced this is a mixed methods study but a multimethod one. It is not indicated in the methods how authors mixed or used findings to complement each other. As indicated in the abstract, how were methods "integrated" exactly? Or, was it a sequential, multi-method approach? Authors would benefit from citing and explaining the use of these two methodologies more clearly.

- The results are long and it is easy to lose the point the authors are trying to make. If the editor so wishes, they could request the authors to pare down the findings section.

- The integration of the quantitative and qualitative findings seem to be slightly superficial in this mixed methods study. There is more to be said about how your findings using each of these methods support each other.

- In the methods, authors indicate a follow up was done in the Spring of 2019. What happened? Why was this done and how did it add value?

- The time gap between data collected and 2019 is not touched on in the discussion about any contextual changes since the data was collected. What do authors think might have changed, or are their findings entirely relevant now?

- I would like to see the authors engage more with the limitations of the quantitative data collected and could these limitations be used to explain your findings?

- The conclusions are particularly effective as they are solutions based.

Reviewer #2: This is a terrific, well-written exploration of race, structural violence and drug selling and using.

Good multi-methodologic studies, like this one, are challenging to execute. The collaboration between disciplines and the requisite conferring on style, language, hypothesis, data and word count are challenging from the start. The word count is justifiably high to allow for adequate explanation of both the quant and qual data. It is the price of getting the right balance in the inter-disciplinary collaboration. I wish there were more papers like this one.

A couple of minor points to address:

- There is some repetition in the simple description of the neighborhood as “impoverished” “drug” “lucrative market” etc.

- Line 575: “Figure 5” might be referring to “6”

- Figure 6 could be explained a bit better.

- The claim made in lines 619-620 does not hold for homicide, with the data presented.

6. PLOS authors have the option to publish the peer review history of their article (what does this mean?). If published, this will include your full peer review and any attached files.

Reviewer #1: No

Reviewer #2: No

---

## [Author Response · Author response to Decision Letter 0]

8 Oct 2019

Please see the attached response to reviewers file, color coded for readability.

---

## [Decision Letter · Decision Letter 1]

5 Nov 2019

Structural vulnerability to narcotics-driven firearm violence: An ethnographic and epidemiological study of Philadelphia’s Puerto Rican inner-city

PONE-D-19-20254R1

Dear Dr. Friedman,

We are pleased to inform you that your manuscript has been judged scientifically suitable for publication and will be formally accepted for publication once it complies with all outstanding technical requirements.

With kind regards,

Cecilia Benoit

Academic Editor

PLOS ONE

Reviewers' comments:

Reviewer's Responses to Questions

**Comments to the Author**

1. If the authors have adequately addressed your comments raised in a previous round of review and you feel that this manuscript is now acceptable for publication, you may indicate that here to bypass the “Comments to the Author” section, enter your conflict of interest statement in the “Confidential to Editor” section, and submit your "Accept" recommendation.

Reviewer #1: All comments have been addressed

Reviewer #2: All comments have been addressed

2. Is the manuscript technically sound, and do the data support the conclusions?

Reviewer #1: Yes

Reviewer #2: Yes

3. Has the statistical analysis been performed appropriately and rigorously? 

Reviewer #1: Yes

Reviewer #2: Yes

4. Have the authors made all data underlying the findings in their manuscript fully available?

Reviewer #1: Yes

Reviewer #2: Yes

5. Is the manuscript presented in an intelligible fashion and written in standard English?

Reviewer #1: Yes

Reviewer #2: Yes

6. Review Comments to the Author

Reviewer #1: Authors have provided a comprehensive response to reviewers and editors comments. This manuscript will make a strong addition to the literature.

Reviewer #2: Thank you for the revisions based on my comments as well as another reviewer and the Editor. I find the revisions satisfactory.

7. PLOS authors have the option to publish the peer review history of their article (what does this mean?). If published, this will include your full peer review and any attached files.

Reviewer #1: No

Reviewer #2: Yes: Dan Ciccarone

---

## [Editor Report · Acceptance letter]

12 Nov 2019

PONE-D-19-20254R1 

Structural vulnerability to narcotics-driven firearm violence: An ethnographic and epidemiological study of Philadelphia’s Puerto Rican inner-city 

Dear Dr. Friedman:

I am pleased to inform you that your manuscript has been deemed suitable for publication in PLOS ONE. Congratulations! Your manuscript is now with our production department. 

With kind regards,

on behalf of

Dr. Cecilia Benoit 

Academic Editor

PLOS ONE